# Anion Effect on Forward Osmosis Performance of Tetrabutylphosphonium-Based Draw Solute Having a Lower Critical Solution Temperature

**DOI:** 10.3390/membranes13020211

**Published:** 2023-02-08

**Authors:** Jihyeon Moon, Hyo Kang

**Affiliations:** BK-21 Four Graduate Program, Department of Chemical Engineering, Dong-A University, 37, Nakdong-Daero 550 Beon-gil, Saha-gu, Busan 49315, Republic of Korea

**Keywords:** ionic liquid, forward osmosis, draw solution, water treatment, lower critical solution temperature

## Abstract

The applicability of ionic liquids (ILs) as the draw solute in a forward osmosis (FO) system was investigated through a study on the effect of the structural change of the anion on the FO performance. This study evaluated ILs composed of tetrabutylphosphonium cation ([P_4444_]^+^) and benzenesulfonate anion ([BS]^−^), para-position alkyl-substituted benzenesulfonate anions (*p*-methylbenzenesulfonate ([MBS]^−^) and *p*-ethylbenzenesulfonate ([EBS^−^]), and methanesulfonate anion ([MS]^−^). The analysis of the thermo-responsive properties suggested that the [P_4444_][MBS] and [P_4444_][EBS] ILs have lower critical solution temperatures (LCSTs), which play a beneficial role in terms of the reusability of the draw solute from the diluted draw solutions after the water permeation process. At 20 wt% of an aqueous solution, the LCSTs of [P_4444_][MBS] and [P_4444_][EBS] were approximately 36 °C and 25 °C, respectively. The water flux and reverse solute flux of the [P_4444_][MBS] aqueous solution with higher osmolality than [P_4444_][EBS] were 7.36 LMH and 5.89 gMH in the active-layer facing the draw solution (AL-DS) mode at osmotic pressure of 25 atm (20 wt% solution), respectively. These results indicate that the [P_4444_]^+^-based ionic structured materials with LCST are practically advantageous for application as draw solutes.

## 1. Introduction

Since water shortages have become a critical global problem owing to the increasing demand for clean water, a supply of adequate water is crucial to sustain public health and national prosperity [1,2,3]; therefore, the development of water treatment technologies is essential. Among the various water purification technologies that have been investigated, forward osmosis (FO) has been considered a feasible energy-efficient process [4,5,6]. Unlike conventional pressure-driven membrane processes, the FO membrane process is activated by the chemical potential difference between a feed solution and draw solution, which are placed on different sides of a semipermeable membrane. This natural phenomenon allows the transport of water molecules across the membrane, leading to the dilution of the draw solution in the FO process [7,8]. FO processes that do not require external pressure exhibit low fouling tendencies and high fouling reversibility of membranes, thereby increasing membrane life [9,10,11]. However, the industry has not fully accommodated the FO process because an additional recovery process to separate water from the diluted draw solution is required, which is energy-intensive [12]. The design or selection of appropriate draw solutions with high water-drawing ability and easy recovery is key in improving the FO process [13,14].

Researchers have proposed two types of draw solutes such as non-responsive [15,16,17] and responsive [18,19,20]. To achieve easy recovery, various responsive draw solutes have been proposed. The responsive draw solutes undergo a change in water solubility upon exposure to external environmental parameters, such as temperature [18,21], pH [22], CO_2_ [23,24], and electric and magnetic fields [25,26,27,28]. Thermo-responsive draw solutes have especially attracted attention for recovering draw solutes and clean water because of their simplicity and low-energy usage in the recovery process via a simple temperature control using waste or geothermal heat [29,30,31]. Thermo-responsive draw solutes can exhibit one of the following two types of phase transition: upper critical solution temperature (UCST) and lower critical solution temperature (LCST). With the UCST-type, the compatibility of the draw solute and water increases upon heating, while the water solubility of the LCST-type draw solute decreases, implying easy recovery.

Various thermo-responsive draw solutes have been explored, which include hydrogels [32,33], homopolymers [18,34], oligomers [20], and copolymers [21]. Recently, thermo-responsive ILs have been considered promising draw solutes because ILs can dissociate in water, leading to osmolality [35,36]. ILs have several distinct characteristics, such as high ionic conductivity, thermal stability, tunable polarity, and negligible volatility [37,38,39,40]. Moreover, the physicochemical properties can be tailored by altering the ion species [41,42]. ILs are defined as compounds composed of a positively charged cation and a negatively charged anion. In the case of cation–anion combinations of ILs that show thermo-responsive phase behavior, typically, the anions include both organic and inorganic species, but cations include usually organic species such as imidazolium, pyridinium, phosphonium, and ammonium [43,44]. Several thermo-responsive ILs were introduced as the draw solute at 20–60 °C, such as tetrabutylammonium 2,4,6-trimethylbenzenesulfonate ([N_4444_]2,4,6-MeBnSO_3_) [45], tetrabutylphosphonium trifluoroacetate ([P_4444_]CF_3_COO) [46], 1-butyl-3-methylimidazolium tetrafluoroborate ([Bmim][BF_4_]) [47], tetraethylammonium bromide ([N_2222_]Br) [48], betaine bis(trifluoromethylsulfonyl)imide ([Hbet][Tf_2_N]) [49], and poly(4-vinylbenzyltributylammonium hexanesulfonate) (P[VBTBA][HS]) [50]. These examples refer to the situation where the thermo-responsive ILs are available for draw solute in the FO system. The relationship between the structure and phase behavior of LCST-type ILs in an aqueous solution has also been investigated. Many reports show that the main factor affecting the LCST behavior of the ILs in water is the total hydrophobicity of the ILs, which can be tuned by changing the hydrophobic moieties of the cation or anion [51,52,53,54,55,56]. Therefore, a structure–property correlation study of ILs can support the design of future draw solutes with water-drawing and simple separation abilities. For example, the effect of the hydrophobic alkyl chain groups on a specific property has been studied using several series of ILs such as quaternary phosphonium-based ILs paired with different anions, including tetrachloroferrate (III), amino acid, and sulfonate types [55,56,57]. Among various ILs, phosphonium-based ILs have been reported to have more thermal and chemical stability [58,59,60]. The phosphonium-based ILs become attractive materials due to their competitive cost of synthesis and higher productivity at the industrial production level as well as their stability [61,62]. According to previous literature, the ILs based on phosphonium with shorter alkyl chain lengths have shown low toxicity toward several human microorganisms and they present the potential for bioprocessing applications [63]. Therefore, we investigated the usability of tetrabutylphosphonium-based ILs in combination with benzenesulfonate which has a sulfonate group attached to the phenyl ring, a simple hydrophobic group. The hydrophobicity of benzenesulfonate anions can be tuned by attaching the extra methyl group to the phenyl ring. In addition, benzenesulfonate also has the advantage of low production cost and high thermal stability [64,65].

This study investigated the water-drawing and separation abilities of a series of tetrabutylphosphonium ([P_4444_]^+^)-based ILs formed by benzenesulfonate anion ([BS]^−^), para-position alkyl-substituted benzenesulfonate anions (*p*-methylbenzenesulfonate ([MBS]^−^) and *p*-ethylbenzenesulfonate ([EBS^−^])), and methanesulfonate anion ([MS]^−^) with different hydrophobicity. The effect of changing the hydrophobic moiety of anions in ILs on their osmolality and LCST behavior was systematically observed. These investigations provide a guide for the development of draw solutes in improving the efficiency of the FO system.

## 2. Materials and Methods

### 2.1. Materials

Tetrabutylphosphonium bromide, [P_4444_]Br (99.0%), sodium benzenesulfonate, Na[BS] (>96.0%), sodium *p*-methylbenzenesulfonate (>90.0%) (Na[MBS]), sodium *p*-ethylbenzenesulfonate, Na[EBS] (>98.0%), and tetrabutylphosphonium methanesulfonate, [P_4444_][MS] (>98.0%) were purchased from Tokyo Chemical Industry Co., Ltd. (Tokyo, Japan). Dichloromethane was purchased from Daejung Chemicals and Metals Co., Ltd. (Sinan, Republic of Korea).

### 2.2. Preparation of Phosphonium-Based ILs

Tetrabutylphosphonium benzenesulfonate ([P_4444_][BS]), tetrabutylphosphonium *p*-methylbenzenesulfonate ([P_4444_][MBS]), and tetrabutylphosphonium *p*-ethylbenzenesulfonate ([P_4444_][EBS]) were prepared by mixing [P_4444_]Br (3.39 g, 10 mmol) and sodium benzene derivatives, Na[BS] (3.60 g, 20 mmol), Na[MBS] (3.88 g, 20 mmol), and Na[EBS] (4.16 g, 20 mmol) in 20 mL of distilled water at 25 °C for 24 h. The products, [P_4444_][BS], [P_4444_][MBS], and [P_4444_][EBS] were extracted with dichloromethane and purified with distilled water. The resulting products were obtained after the removal of dichloromethane by evaporation and drying in a vacuum oven at 40 °C for 48 h. In contrast, tetrabutylphosphonium methanesulfonate ([P_4444_][MS]) was used as received without any further processing. To identify the structure of the ILs, proton nuclear magnetic resonance (^1^H-NMR) spectroscopy was used.

^1^H-NMR of [P_4444_][BS] (400 MHz, D_2_O, *δ*/ppm): *δ* = 0.84–1.01 (t, 12H, -P^+^-CH_2_-CH_2_-CH_2_-*CH_3_*), 1.35–1.66 (m, 16H, -P^+^-CH_2_-*CH_2_*-*CH_2_*-CH_3_), 2.03–2.28 (t, 8H, -P^+^-*CH_2_*-CH_2_-CH_2_-CH_3_, 7.46–7.69, 7.72–7.92 (s, 5H, *PhH*-SO_3_^−^).

^1^H-NMR of [P_4444_][MBS] (400 MHz, D_2_O, *δ*/ppm): *δ =* 0.83–1.03 (t, 12H, -P^+^-CH_2_-CH_2_-CH_2_-*CH_3_*), 1.35–1.67 (m, 16H, -P^+^-CH_2_-*CH_2_*-*CH_2_*-CH_3_), 2.06–2.25 (m, 3H, -*CH_3_*-Ph-SO_3_^−^), 2.31–2.49 (t, 8H, -P^+^-*CH_2_*-CH_2_-CH_2_-CH_3_), 7.29–7.45, 7.61–7.77 (s, 4H, CH_3_-*PhH*-SO_3_^−^).

^1^H-NMR of [P_4444_][EBS] (400 MHz, D_2_O, *δ*/ppm): *δ* = 0.72–1.07 (t, 12H, -P^+^-CH_2_-CH_2_-CH_2_-*CH_3_*), 1.12–1.36 (t, 3H, -*CH_3_*-CH_2_-Ph-), 1.36–1.83 (m, 16H, -P^+^-CH_2_-*CH_2_*-*CH_2_*-CH_3_), 1.90–2.51 (t, 8H, -P^+^-*CH_2_*-CH_2_-CH_2_-CH_3_), 2.61–2.88 (m, 2H, -CH_3_-*CH_2_*-Ph-), 7.30–7.52 (d, 4H, CH_3_-CH_2_-*PhH*-SO_3_^−^), 7.62–7.81 (d, 4H, CH_3_-CH_2_-*PhH*-SO_3_^−^).

^1^H-NMR of [P_4444_][MS] (400 MHz, D_2_O, *δ*/ppm): *δ* = 0.79–1.07 (t, 12H, -P^+^-CH_2_-CH_2_-CH_2_-*CH_3_*), 1.29–1.75 (m, 16H, -P^+^-CH_2_-*CH_2_*-*CH_2_*-CH_3_), 1.99–2.31 (t, 8H, -P^+^-*CH_2_*-CH_2_-CH_2_-CH_3_), 2.70–2.90 (s, 3H, *CH_3_*-SO_3_-).

### 2.3. FO Performance

The water flux, which represents the FO performance, was measured using a lab-scale FO apparatus connected to custom-made, L-shaped glass tubes. The membrane (Hydration Technologies Inc. (HTI, Albany, OR, USA), commercial thin film composite membrane) was inserted between two glass tubes composed of a 3.3 cm diameter fastening aluminum clamp and FO was carried out to evaluate its performance. The parameter of the FO performance or water flux (*J_w_*) was measured in the active-layer facing the draw solution (AL-DS) mode and the active-layer facing the feed solution (AL-FS) mode, respectively. Distilled water was added as a feed solution to one of the tubes and [P_4444_][MBS] was added as a draw solution to the other. The measurement was performed by stirring with a solenoid (OCTOPUS CS-4, AS ONE, Osaka, Japan) at 25 ± 1 °C for 20 min.

We calculated the water flux, *J*_w_, by converting the height difference to the volumetric change of the draw solution using the following Equation (1):(1)$$J_{w}=\frac{\Delta V}{A\Delta t}$$
where Δt [h] is the handling time of the experiment of the FO process, ΔV [L] is the variation in volume of the draw solution, and $A\;\left[m^{2}\right]$ is the effective membrane area, which is $4.15\times 10^{-4}\;m^{2}$.

The reverse solute flux (*J_s_*, g m^–2^ h^–1^, gMH) represents the quantity of the permeated draw solute across the FO membrane to the feed solution. We calculated the reverse solute flux by comparing the conductivity difference of the feed solution before and after FO using the following Equation (2):(2)$$J_{s}=\frac{\Delta \left(\mathrm{CV}\right)}{A\Delta t}$$
where ΔC [mg/L] is the concentration change and ΔV [L] is the variation in volume of the feed solution before and after FO.

### 2.4. Instruments

To confirm the molecular structure of the prepared ILs, ^1^H-NMR spectroscopy (MR400 DD2 NMR, Agilent Technologies, Inc., Santa Clara, CA, USA) and Fourier transform infrared (FT-IR) spectroscopy (Nicolet iS20, Thermo Fisher Scientific Inc., Waltham, MA, USA) were used. A conductivity meter (Seven2CO pro, METTLER TOLEDO Inc., Columbus, OH, USA) was used to measure the conductivity. An osmometer (SEMI-MICRO OSMOMETER K-7400, KNAUER Wissenschaftliche Geräte GmbH Co., Berlin, Germany) was used to obtain the osmotic pressure of the IL aqueous solutions. An ultraviolet-visible (UV-Vis) spectrophotometer (EMC-11D-V, EMCLAB Instruments GmbH Co., Duisburg, Germany) fitted with a temperature controller (TC200P, Misung Scientific Co., Ltd., Yangju, Republic of Korea) was used to confirm the phase separation temperature of the ILs in water. The measurement of the contact angle was performed using a Krüss DSA10 (KRÜSS Scientific Instruments Inc., Hamburg, Germany) contact angle analyzer equipped with drop shape analysis software after deposing the water and aqueous solution of IL droplets on the FO membrane surface. The average volume of the droplets was 5 μL. The contact angles of each solution were determined four or more times.

## 3. Results and Discussion

### 3.1. Synthesis and Characterization of Phosphonium-Based ILs

The tetrabutylphosphonium-based ILs with benzenesulfonate derivative anions were prepared via anion exchange of tetrabutylphosphonium bromide ([P_4444_]Br) from bromide to [BS], [MBS], and [EBS], as illustrated in Figure 1. The structures of [P_4444_][BS], [P_4444_][MBS], and [P_4444_][EBS] were determined using ^1^H-NMR and attenuated total reflectance FT-IR (ATR FT-IR) analysis. The ^1^H-NMR spectra of the [P_4444_][BS], [P_4444_][MBS], [P_4444_][EBS], and [P_4444_][MS] are depicted in Figure 2a–d. The successful preparation of [P_4444_][BS], [P_4444_][MBS], [P_4444_][EBS], and [P_4444_][MS] was confirmed by calculating the integral ratio for each region of the protons of the alkyl groups. Furthermore, the anion exchange of [P_4444_][BS], [P_4444_][MBS], and [P_4444_][EBS] was confirmed by comparison of the respective peaks of the resulting products with those of Na[BS], Na[MBS], and Na[EBS] using FT-IR spectroscopy, as illustrated in Figure 3a–c. [P_4444_][MS] was also confirmed (Figure 3d). All resulting products exhibit two characteristic peaks corresponding to asymmetrical and symmetrical stretching vibrations of C–H in the alkyl groups at 2962–2957 cm^−1^ and 2870–2867 cm^−1^, respectively. Although the absorption at 1600–1500 cm^−1^ should be assigned to the stretching of the C=C bands in the benzene ring, the coarse spectra make it difficult to trace the variation in this region [66]. However, bands at 1200–1193 cm^−1^ and 1120–1115 cm^−1^ correspond to the asymmetrical and symmetrical stretching of S=O, which makes it possible to follow the anion variations. Therefore, the absorption of C–H and S=O bands brings particular convenience to the analysis of cations and anions in this system.

### 3.2. Electrical Conductivity

As an indicator of the ion dissociation ability, electrical conductivity is affected by the ionic species, which influence the charge carrier and ion mobility. Typically, the large ions cause ion aggregation/pairing and have more interfacial area between the ion and solvent, increasing the hydrodynamic resistance, and resulting in limited ion mobility [67,68]. In addition, the conductivity is related to the osmolality of ILs, which is largely dependent on ion dissociation [69]. Thus, the conductivities of the [P_4444_][MS], [P_4444_][BS], [P_4444_][MBS], and [P_4444_][EBS] aqueous solutions were measured at the concentration range of 5–20 wt%. The conductivities of the [P_4444_][MS], [P_4444_][BS], [P_4444_][MBS], and [P_4444_][EBS] aqueous solutions were approximately 9187, 5460, 4510, and 3710 μS/cm, respectively, at a concentration of 10 wt%. The conductivities of [P_4444_][MS], [P_4444_][BS], [P_4444_][MBS], and [P_4444_][EBS] aqueous solutions increased to approximately 12,900, 7037, 5436, and 4009 μS/cm, respectively, when their concentration was increased to 20 wt%. As shown in Figure 4, the conductivities of the IL aqueous solutions increase with increasing concentration owing to increasing numbers of mobile ions in the aqueous solution in the range of 5–20 wt% [70,71,72,73,74]. Furthermore, the conductivity decreases in the order [P_4444_][MS] < [P_4444_][BS] < [P_4444_][MBS] < [P_4444_][EBS], which is also the order of increasing ion size at any concentration of ILs. As aforementioned, the charge carriers and ionic mobility of the ILs containing large ion sizes decrease, resulting in a decrease in ionic conductivity. In addition, aromatic substituents enhance the van der Waals interactions, which lead to low ion conductivities [75]. Therefore, tetrabutylphosphonium-based ILs containing small anions show a relatively high conductivity.

### 3.3. Osmotic Pressure

The transport of the water molecules across the membrane is activated by the high osmotic pressure of the draw solution compared to that of the feed solution; thus, it can be used as the driving force of the FO system [30,76]. As a colligative property, the increase in the concentration of the draw solution enhances its osmotic pressure [77]. To investigate the applicability as a draw solute, the osmolality values of the [P_4444_][MS], [P_4444_][BS], [P_4444_][MBS], and [P_4444_][EBS] aqueous solutions were measured at increasing IL concentrations from 5 to 20 wt% using the freezing point depression method. The osmolality values could be converted to the osmotic pressure via the van’t Hoff equation using the temperature (T = 297 K) and density of the solution (density = 1 g/mL), as depicted in Figure 5. The osmotic pressures of the [P_4444_][MS], [P_4444_][BS], [P_4444_][MBS], and [P_4444_][EBS] aqueous solutions increased from 17, 14, 11, and 9 atm to 34, 33, 25, and 18 atm, respectively, when their concentration increased from 10 to 20 wt%. As a colligative property, the osmotic pressure is correlated with the degree of ion dissociation in water, which is largely dependent on ion solvation [78,79]. In addition to cation–anion interactions, solvent–ion, and solvent–solvent interactions determine how well the ions are solvated, and consequently the degree of ion dissociation [80]. Hydrogen bonding, particularly between solvent and anion, plays a key role in ion solvation [81]. The extent of ion dissociation decreases with increasing hydrophobicity of the anion owing to the increased size of the hydrophobic moiety such as a benzene derivative; this requires a greater number of water molecules for solvation based on anion size [82,83,84]. Thus, ILs with highly hydrophilic [P_4444_][MS], containing [MS]^–^ anions, which is considered a factor in their high degree of ion dissociation, show a higher osmotic pressure at all concentrations. Moreover, the addition of the hydrophobic methyl group to the benzenesulfonate anion decreases the solvation of the benzenesulfonate-based ILs with water; thus, the osmotic pressure decreases in the order: [P_4444_][BS] > [P_4444_][MBS] > [P_4444_][EBS]. Therefore, at any concentration of the ILs, the osmotic pressure increases in the order of [P_4444_][MS] > [P_4444_][BS] > [P_4444_][MBS] > [P_4444_][EBS], which is also the order of increasing hydrophilicity. The [P_4444_][MS], [P_4444_][BS], and [P_4444_][MBS] aqueous solutions generate osmotic pressures of 34–25 atm at a concentration of 20 wt%, as compared to 26 atm generated by traditional inorganic salts especially NaCl with 3.5 wt%. Although the osmotic pressures of the [P_4444_]-based ILs aqueous solutions are lower than NaCl, they exhibit potential as draw solution for brackish water treatment and food processing [85]. In addition, overall, the [P_4444_]-based ILs have not only good solubility and a high degree of dissociation at a concentration of 20 wt% but also thermal recovery properties.

### 3.4. Thermo-Responsive Property

Thermo-responsive ILs facilitate the efficient separation from the diluted draw solution into clean water and draw solute [86]. The LCST is the phase separation temperature at which the ILs with LCST become a heterogeneous state in an aqueous solution above the LCST. The LCST behavior is attributed to the following equation, where Δ*G_mix_* is the mixing free energy, Δ*H_mix_* is the mixing enthalpy, and Δ*S_mix_* is the mixing entropy [87,88,89].
Δ*G_mix_* = Δ*H_mix_* − *T*Δ*S_mix_*
(3)

In the aqueous solution of an IL system, the interaction between ions and water, such as hydrogen bonding plays an important role in the LCST-type phase behavior of the IL. The mixing enthalpy is negative at low temperature because the hydrogen bonding makes a negative Δ*S_mix_* and a miscible phase between the component species, thus Δ*G_mix_* must be negative. Upon heating to a temperature above LCST, Δ*G_mix_* changes from negative to positive leading to phase separation due to the breaking of the hydrogen bonding. In our system, the [P_4444_][MBS] and [P_4444_][EBS] aqueous solutions exhibit an LCST-type phase transition, as can be seen in Figure 6. Below its LCST, the hydrogen bonding interaction between water and the [P_4444_][MBS] and [P_4444_][EBS] is dominant. But above its LCST, the hydrogen bonding interaction is weakened and the hydrophobic interaction between [P_4444_]^+^ and benzenesulfonate derivatives anions ([MBS]^–^ and [EBS]^–^) becomes dominant, thereby inducing the phase separation of [P_4444_][MBS] and [P_4444_][EBS]. In contrast, there were no noticeable changes in transmittance for the [P_4444_][MS] and [P_4444_][BS] aqueous solutions within the entire analyzed temperature range. Because the main factor affecting the LCST behavior of the ILs in water is the total hydrophobicity of the ILs [56]. When the IL composed of cation and anion is highly hydrophilic, IL is miscible in water and does not exhibit phase separation. The LCST of the [P_4444_][MBS] and [P_4444_][EBS] aqueous solutions were determined by UV-Vis spectroscopy and temperature controller at 550 nm upon heating to 90 °C. The transmittance versus temperature curves of the [P_4444_][MBS] and [P_4444_][EBS] aqueous solutions mark changes at each LCST for concentrations of 10, 15, and 20 wt%, as shown in Figure 6. As the concentration of the [P_4444_][MBS] and [P_4444_][EBS] aqueous solution increases, the LCST decreases. The LCSTs of the [P_4444_][MBS] aqueous solutions are approximately 54, 39, and 36 °C, at each concentration, respectively. Additionally, the respective LCSTs of [P_4444_][EBS] are approximately 31, 28, and 19 °C. As a result of the thermo-responsive phase separation behavior of the prepared ILs in an aqueous solution, the [P_4444_][MBS] and [P_4444_][EBS] aqueous solutions could be easily separated from the draw solute and pure water by heating and cooling, requiring minimal recovery energy during FO owing to their thermo-responsiveness.

### 3.5. Contact Angle of [P_4444_][MBS] on the FO Membrane

The wettability of the membrane surface is a macroscopic representation of the interfacial interaction between fluid and membrane surface and plays an important role in the membrane performance [90,91,92,93]. The surface wetting of the membrane is usually characterized by the contact angle. The contact angles of the [P_4444_][MBS] aqueous solutions with various concentrations and distilled water on the surface of the active layer in the FO membrane were measured to understand the effect of the draw solution ([P_4444_][MBS] aqueous solution) on the wettability of the membrane. As shown in Figure 7, the average contact angle of distilled water is lower than that of [P_4444_][MBS] aqueous solution at all concentrations on the membrane surface and follows the following trend: distilled water > 5 wt% [P_4444_][MBS] > 10 wt% [P_4444_][MBS] > 15 wt% [P_4444_][MBS] > 20 wt% [P_4444_][MBS]. The distilled water contact angle was 75°. The contact angles of the [P_4444_][MBS] aqueous solutions were 54, 49, 39, and 21°, respectively, at concentrations of 5, 10, 15, and 20 wt%, respectively. The contact angle of [P_4444_][MBS] aqueous solution on the membrane becomes smaller with increasing concentration. This means that the wettability of the FO membrane was enhanced on decreasing the contact angle of the [P_4444_][MBS] aqueous solution from 54° to 21°.

### 3.6. Water Flux (J_w_) and Reverse Solute Flux (J_s_) of [P_4444_][MBS]

The water flux and reverse solute flux of [P_4444_][MBS], which has a high osmolality potential among the thermo-responsive ILs, were measured to evaluate [P_4444_][MBS] as a draw solute at different orientations of the membrane placed between the two connected glass tubes; the glass tubes were filled with distilled water and [P_4444_][MBS] aqueous solution, respectively. At 25 ± 1 °C, the water fluxes and reverse solute fluxes of the [P_4444_][MBS] aqueous solutions were measured in the AL-DS and AL-FS modes when the concentration of ILs varied from 5 to 20 wt%. The concentrations of 5, 10, 15, and 20 wt% are represented as 5, 11, 17, and 25 atm, respectively. The water fluxes of the [P_4444_][MBS] aqueous solutions were 1.67, 3.27, 4.64, and 7.36 LMH in the AL-DS mode at osmotic pressures of 5, 11, 17, and 25 atm, respectively, and 1.65, 2.80, 3.45, and 4.85 LMH in the AL-FS mode, respectively. The reverse solute flux is the quantity of the permeated draw solute across the FO membrane to the feed water and was calculated by comparing the amount of total dissolved solids (TDS) of the distilled water (feed solution) before and after FO [94]. The reverse solute fluxes of the [P_4444_][MBS] aqueous solutions were 1.36, 1.47, 3.77, and 5.89 gMH in the AL-DS mode at 5, 11, 17, and 25 atm, respectively, and 1.14, 1.25, 2.45, and 2.67 gMH in the AL-FS mode, respectively, for the same condition. We observed that the water flux increases when the osmotic pressure of IL increases. This trend indicates that the osmotic pressure induces water permeation for the FO process. Figure 8 shows that the water fluxes are higher for the AL-DS mode than for the AL-FS mode at all [P_4444_][MBS] aqueous solutions, which is attributed to a significant dilutive internal concentration polarization (ICP) effect in the AL-FS mode [95]. This causes the dilution of the draw solution, which reduces the effective osmotic pressure across the membrane. When the feed solution is distilled water, the ICP effect is negligible in the AL-DS mode [96,97].

### 3.7. Recyclability Study of [P_4444_][MBS]

To explore the recyclable property of the [P_4444_][MBS], the FO performance of 20 wt% aqueous solutions of the [P_4444_][MBS] was repeated four times when distilled water was used as the feed solution. The thermal treatment method was used to obtain the recycled [P_4444_][MBS]. As shown in Figure 9a,b, the osmotic pressure and LCST of the [P_4444_][MBS] were measured to confirm the recyclability of [P_4444_][MBS] at the fourth run. The osmotic pressure values at the 4th run were almost the same as that of pristine [P_4444_][MBS], while the LCST value slightly increased after the 4th run. These recycling results clearly show that [P_4444_][MBS] can be easily recycled with relatively low energy consumption without significant loss.

## 4. Conclusions

The water-drawing ability and LCST-type phase transition behavior of IL draw solutes composed of tetrabutylphosphonium cation ([P_4444_]^+^) and anions with different hydrophobicity ([BS]^−^, [MBS]^−^, [EBS]^−^, and [MS]^−^) were described in this article. The total hydrophobicity of the ILs affects the osmolality and LCST-type phase behavior of the draw solution. The hydrophilicity of the draw solutions followed the order [P_4444_][MS] > [P_4444_][BS] > [P_4444_][MBS] > [P_4444_][EMS] at all concentrations. ILs with high hydrophilicity have a high osmolality potential and LCST. During the phase transition behavior, LCSTs were observed from the [P_4444_][MBS] and [P_4444_][EBS] aqueous solutions. The LCSTs of the 20 wt% [P_4444_][MBS] and [P_4444_][EBS] aqueous solutions were approximately 36 °C and 25 °C, respectively. In the FO performance test, at an osmotic pressure of 25 atm, the water fluxes and reverse solute fluxes of the thermo-responsive [P_4444_][MBS] aqueous solution that exhibited higher osmolality compared to [P_4444_][EBS] were 7.36 LMH and 5.89 gMH in the AL-DS, and 4.85 LMH and 2.67 gMH in the AL-FS mode at 25 ± 1 °C. Therefore, these results are an important reference for the development of improved draw solutions in the FO system.

## Figures and Tables

**Figure 1 membranes-13-00211-f001:**
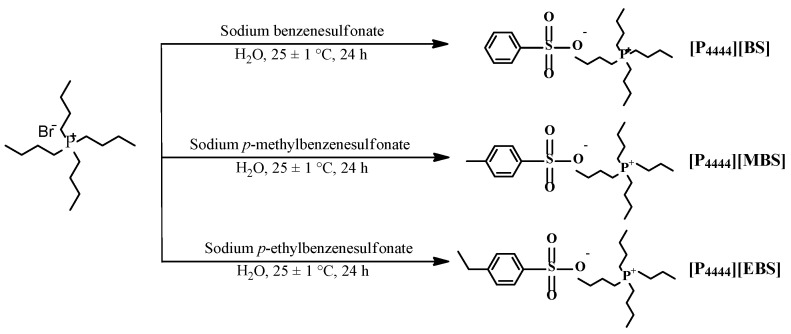
Preparative scheme for [P_4444_][BS], [P_4444_][MBS], and [P_4444_][EBS].

**Figure 2 membranes-13-00211-f002:**
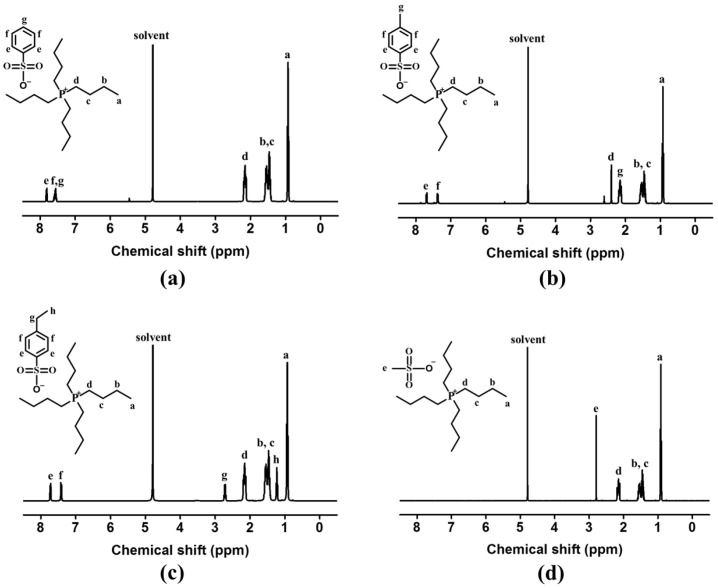
Proton nuclear magnetic resonance (^1^H-NMR) spectra of (**a**) [P_4444_][BS], (**b**) [P_4444_][MBS], (**c**) [P_4444_][EBS], and (**d**) [P_4444_][MS].

**Figure 3 membranes-13-00211-f003:**
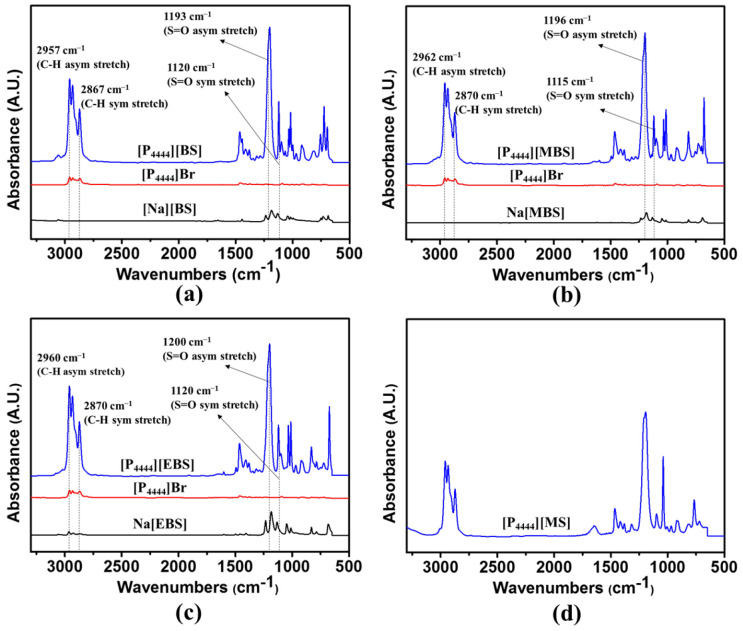
Fourier-transform infrared (FT-IR) spectra of (**a**) [P_4444_][BS], (**b**) [P_4444_][MBS], (**c**) [P_4444_][EBS], and (**d**) [P_4444_][MS].

**Figure 4 membranes-13-00211-f004:**
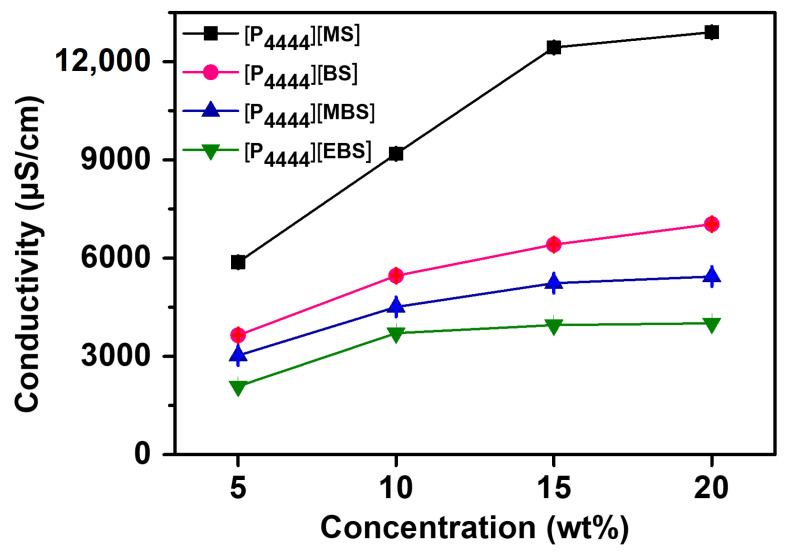
Electrical conductivities of [P_4444_][MS], [P_4444_][BS], [P_4444_][MBS], and [P_4444_][EBS] aqueous solutions according to the concentration of solution.

**Figure 5 membranes-13-00211-f005:**
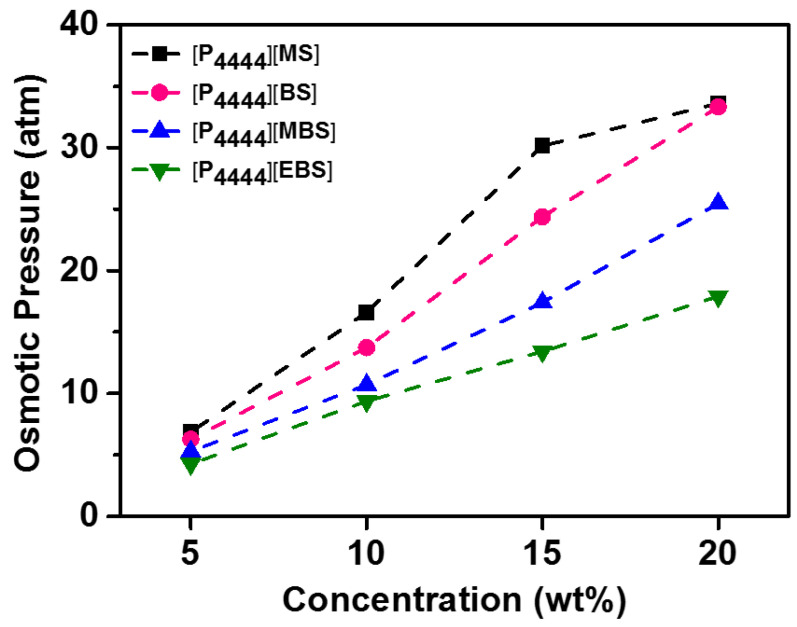
Osmotic pressures of [P_4444_][MS], [P_4444_][BS], [P_4444_][MBS], and [P_4444_][EBS] aqueous solutions according to the concentration of solution.

**Figure 6 membranes-13-00211-f006:**
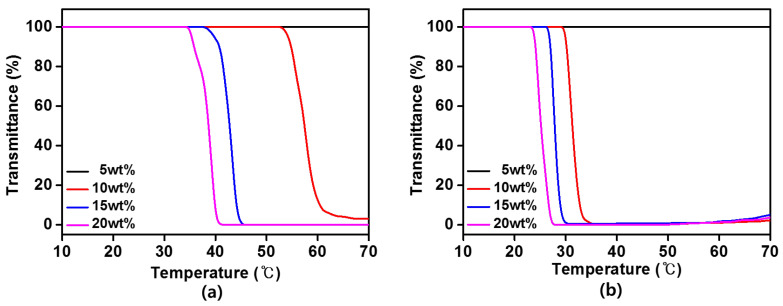
Transmittance curve of (**a**) [P_4444_][MBS] and (**b**) [P_4444_][EBS] aqueous solutions according to the temperature.

**Figure 7 membranes-13-00211-f007:**
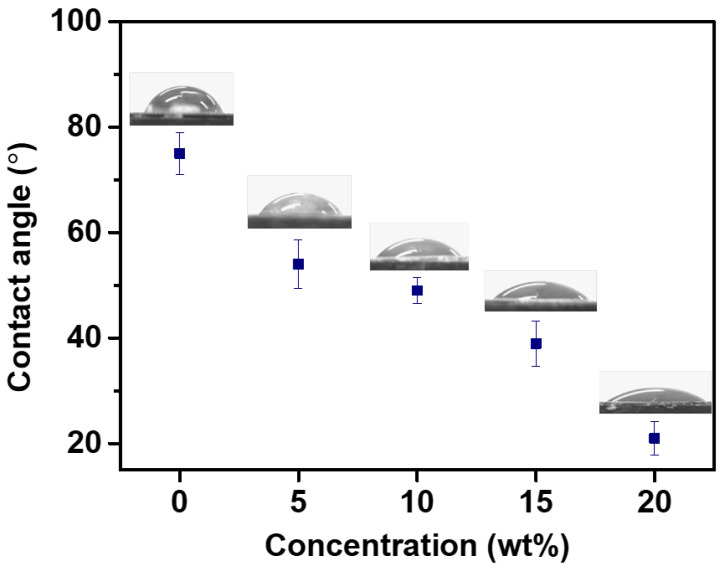
Contact angles of [P_4444_][MBS] aqueous solution at concentrations of 0 (distilled water), 5, 10, 15, and 20 wt% on the FO membrane.

**Figure 8 membranes-13-00211-f008:**
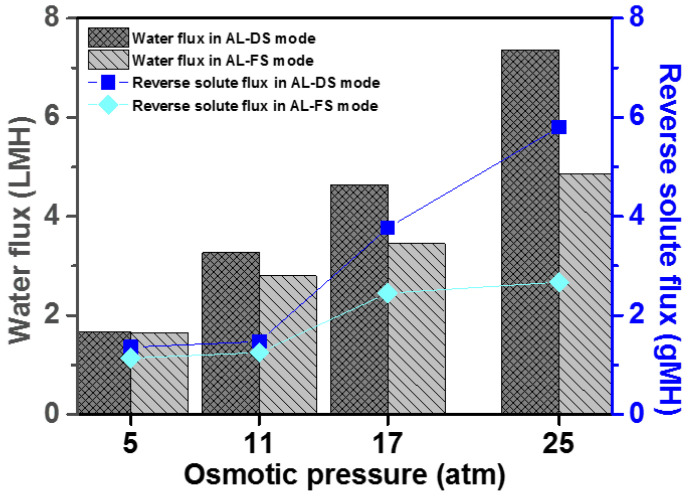
Water fluxes and reverse solute fluxes of [P_4444_][MBS] aqueous solutions according to the osmotic pressure in AL-DS mode and AL-FS mode at a temperature of 25 ± 1 °C during FO process.

**Figure 9 membranes-13-00211-f009:**
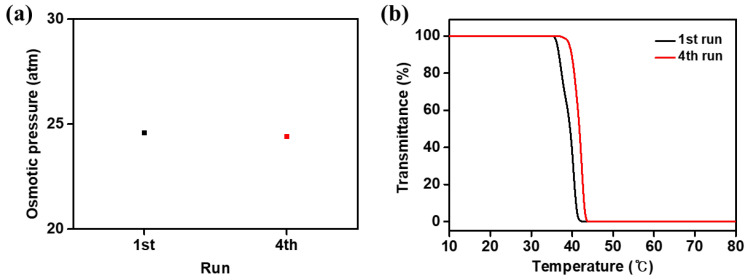
(**a**) Osmotic pressures and (**b**) LCST results using 20 wt% [P_4444_][MBS] aqueous solution as a draw solution and distilled water as a feed solution. From the 1st to the 4th run, the recovered [P_4444_][MBS] from the previous run was used.

## Data Availability

The data presented in this study are available on request from the corresponding author.

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
