# Peer review of "Anion Effect on Forward Osmosis Performance of Tetrabutylphosphonium-Based Draw Solute Having a Lower Critical Solution Temperature"

_membranes, 2023, doi:10.3390/membranes13020211_

Round 1

Reviewer 1 Report

This manuscript describes the preparation of three phosphonium-based ionic liquids as a draw solute for forward osmosis. The difference between the structure of the anion groups of the ILs and its influence on FO performance was explored. The manuscript is quite straightforward and can potentially be useful; however, the manuscript fails to present its novelty and importance. The results presented in this work are not unprecedented, and are easily overperformed by other materials. Also, the discussion of the results feels lacking, and can be supplemented further. There were also already previous works done on phosphonium-based ILs (cited in this work) and this work does not set itself apart from those works.  I therefore recommend that this work be accepted for publication in Membranes after major revision.

Specific comments:

1.       Title: Can be reorganized for better understanding; can be confusing for readers

2.       Introduction: Can further be expanded to introduce more previous work on IL as draw solute for FO

3.       H-NMR data were presented redundantly: Fig 2, Materials and method section, and Results and discussion section

4.       Fig 7 can be presented as water flux vs osmotic pressure difference

Author Response

Dear Editor at Membranes

We gratefully appreciate your kind reviewing and considering for publication in “Membranes”. We are submitting a revised manuscript (Membranes-2193805) entitled “Anion Effect on Forward Osmosis Performance of Tetrabutylphosphonium-based Draw Solute Having Lower Critical Solution Temperature”

We carefully read the reviewer’s comments and your e-mail. Reviewers gave us helpful comments on our manuscript. We think the reviewer’s opinion and suggestion is fairly reasonable. Therefore, we revised our manuscript taking the reviewer’s comments into consideration as follows. As you and the Reviewer suggested we modified some parts of the manuscript and the changes are shown in yellow texts. These changes are listed as follows:

Referee’s comments:

Referee: 1

Comments: This manuscript describes the preparation of three phosphonium-based ionic liquids as a draw solute for forward osmosis. The difference between the structure of the anion groups of the ILs and its influence on FO performance was explored. The manuscript is quite straightforward and can potentially be useful; however, the manuscript fails to present its novelty and importance. The results presented in this work are not unprecedented, and are easily overperformed by other materials. Also, the discussion of the results feels lacking, and can be supplemented further. There were also already previous works done on phosphonium-based ILs (cited in this work) and this work does not set itself apart from those works. I therefore recommend that this work be accepted for publication in Membranes after major revision.

Specific comments:

  1. Title: Can be reorganized for better understanding; can be confusing for readers

Answer:

We deeply thank you for the comment suggested by Reviewer and we revised the title to “Anion Effect on Forward Osmosis Performance of Tetrabutylphosphonium-based Draw Solute Having Lower Critical Solution Temperature”. Reviewer’s comments were very helpful to improve the quality of our manuscript.

  1. Introduction: Can further be expanded to introduce more previous work on IL as draw solute for FO

Answer:

We deeply thank you for the comment suggested by Reviewer and we added the text and literatures about previously reported IL as draw solute for FO in Introduction section as, “ILs are defined as compounds composed of a positively charged cation and a negatively charged anion. In the case of cation-anion combinations of ILs that show thermo-responsive phase behavior, typically, the anions include both organic and inorganic species, but cations include usually organic species such as imidazolium, pyridinium, phosphonium, and ammonium [43,44]. Several thermo-responsive ILs were introduced as the draw solute with 20-60 °C, such as tetrabutylammonium 2,4,6-trimethylbenzenesulfonate ([N4444]2,4,6-MeBnSO3) [45], tetrabutylphosphonium trifluoroacetate ([P4444]CF3COO) [46], 1-butyl-3-methylimidazolium tetrafluoroborate ([Bmim][BF4]) [47], tetraethylammonium bromide ([N2222]Br) [48], betaine bis(trifluoromethylsulfonyl)imide ([Hbet][Tf2N]) [49], and poly(4-vinylbenzyltributylammonium hexanesulfonate) (P[VBTBA][HS]) [50]. These examples refer that the thermo-responsive ILs are available for draw solute in the FO system.” We strongly believe that Reviewer will satisfy our answer to this question.

[43] Abdullah, M.; Man, M.S.; Abdullah, S.B.; Saufi, S.M. Synthesis and Characterization of Thermo-Responsive Ionic Liquids (TRILs). IOP Conf. Ser.: Mater. Sci. Eng. 2020, 736, 042027.

[44] Qiao, Y.; Ma, W.; Theyssen, N.; Chen, C.; Hou, Z. Temperature-Responsive Ionic Liquids: Fundamental Behaviors and Catalytic Applications. Chem. Rev. 2017, 117, 6881–6928.

[45] Zeweldi, H.G.; Bendoy, A.P.; Park, M.J.; Shon, H.K.; Kim, H.; Johnson, E.M.; Kim, H.; Lee, S.; Chung, W.; Nisola, G.M. Tetrabutylammonium 2, 4, 6-Trimethylbenzenesulfonate as an Effective and Regenerable Thermo-Responsive Ionic Liquid Drawing Agent in Forward Osmosis for Seawater Desalination. Desalination 2020, 495, 114635.

[46] Nyan, P.S.; Saufi, S.M.; Abdullah, S.B.; Seman, M.N.A.; Taib, M.M. Tetrabutylphosphonium Trifluoroacetate ([P4444] CF3COO) Thermoresponsive Ionic Liquid as a Draw Solution for Forward Osmosis Process. Malaysian J. Anal. Sci. 2018, 22, 605–611.

[47] Abdullah, M.A.M.; Seman, M.N.A.; Chik, S.M.S.T.; Abdullah, S.B. Factorial Design in Optimizing Parameters for Thermoresponsive Ionic Liquids as Draw Solution. Process Saf. Environ. Prot. 2022, 161, 34–49.

[48] Zeweldi, H.G.; Limjuco, L.A.; Bendoy, A.P.; Kim, H.; Park, M.J.; Shon, H.K.; Johnson, E.M.; Lee, H.; Chung, W.; Nisola, G.M. The Potential of Monocationic Imidazolium-, Phosphonium-, and Ammonium-Based Hydrophilic Ionic Liquids as Draw Solutes for Forward Osmosis. Desalination 2018, 444, 94–106.

[49] Zhong, Y.; Feng, X.; Chen, W.; Wang, X.; Huang, K.; Gnanou, Y.; Lai, Z. Using UCST Ionic Liquid as a Draw Solute in Forward Osmosis to Treat High-Salinity Water. Environ. Sci. Technol. 2016, 50, 1039–1045.

[50] Ju, C.; Park, C.; Kim, T.; Kang, S.; Kang, H. Thermo-Responsive Draw Solute for Forward Osmosis Process; Poly (Ionic Liquid) having Lower Critical Solution Temperature Characteristics. RSC adv. 2019, 9, 29493–29501.

  1. H-NMR data were presented redundantly: Fig 2, Materials and method section, and Results and discussion section

Answer:

We deeply thank you for the comment suggested by Reviewer and we revised the text about 1H-NMR data as “The 1H-NMR spectra of the [P4444][BS], [P4444][MBS], [P4444][EBS], and [P4444][MS] are depicted in Fig. 2 (a)–(d). The successful preparation of [P4444][BS], [P4444][MBS], [P4444][EBS], and [P4444][MS] is confirmed by calculating the integral ratio for each region of the protons of the alkyl groups.” in Results and Discussion section. We strongly believe that Reviewer will satisfy our answer to this question.

  1. Fig 7 can be presented as water flux vs osmotic pressure difference

Answer:

We deeply thank you for the comment suggested by Reviewer and we revised this point as “We represented the concentration of 5, 10, 15, and 20 wt% as 5, 11, 17, and 25 atm, respectively. The water fluxes of the [P4444][MBS] aqueous solutions were 1.67, 3.27, 4.64, and 7.36 LMH in the AL-DS mode at osmotic pressure of 5, 11, 17, and 25 atm, respectively, and 1.65, 2.80, 3.45, and 4.85 LMH in the AL-FS mode, respectively.” in Results and Discussion section. We also revised Figure 8. The revised Figure is below. We strongly believe that Reviewer will satisfy our answer to this question.

Figure 8. Water Fluxes and reverse solute fluxes of [P4444][MBS] aqueous solutions according to the osmotic pressure in AL-DS mode and AL-FS mode at a temperature of 25±1 oC during FO process.

We believe that now we answered all of the comments pointed out by the reviewers. I hope that now this paper is publishable in “Membranes”, one of the top journals in the field of membrane applications.

We also believe that this paper is also suitable for publication in “Membranes” from the following reasons.

  1. We prepared tetrabutylphosphonium-based ionic liquids (ILs), tetrabutylphosphonium benzenesulfonate ([P4444][BS]), tetrabutylphosphonium p-methylbenzenesulfonate ([P4444][MBS]), tetrabutylphosphonium p-ethylbenzenesulfonate ([P4444][EBS]), and tetrabutylphosphonium methanesulfonate ([P4444][MS]) to investigate their applicability as draw solute in forward osmosis (FO) process.

  1. In aqueous solutions, [P4444][MBS] and [P4444][EBS] show lower critical solution temperature (LCST) characteristics, which is essential for recovery of the draw solute after the FO permeation process. The LCSTs of 20 wt% aqueous solutions of [P4444][MBS] and [P4444][EBS] were approximately 36 and 25 ºC, respectively. These results signify that [P4444][MBS] and [P4444][EBS] aqueous solutions are efficient in terms of energy requirements for recovering the draw solute after FO process.

  1. The structure-property relationship is very important for designing the suitability of thermo-responsive ILs for draw solutions as well as the design of materials. Because LCSTs of thermo-responsive ILs affected by hydrophilicity and hydrophobicity of the cation or anion species are distinct and attractive characteristics in a narrow range between miscible and immiscible properties in solution. Therefore, we investigate the effect of change in the hydrophobic moiety of anions in ILs on the FO permeation and recovery behavior by varying anions with different hydrophobicity (benzenesulfonate anion ([BS]), para-position alkyl-substituted benzenesulfonate anions (p-methylbenzenesulfonate ([MBS]) and p-ethylbenzenesulfonate ([EBS])), and methanesulfonate anion ([MS])) in tetrabutylphosphonium ([P4444]+)-based ILs to design appropriate draw solutes. As a result, this study could contribute to the basic understanding of how the structure of the ions influences the drawing ability and recovery property of the ILs.

  1. We believe that our study makes a significant contribution to the literature because the thermo-responsive IL, which provides the thermo-responsive liquid-liquid phase separation property in the aqueous solutions was shown to be a promising draw solute that may breakthroughs, surpassing the limitations of technologies in existence such as high recovery temperature such as LCST, and trade-off relationship between osmotic pressure and the recovery temperature in the ionic liquid system.

In view of these achievements, we believe that our work represents a timely methodological advance and breakthrough in the field of membrane applications and thus is appropriate for a journal with the scope and wider readership of “Membranes”.

I hereby certify that this manuscript consists of original, unpublished work which is not under consideration for publication elsewhere.

We are excited to share our manuscript with you and look forward to hearing good news from you soon.

Thank you very much for your time and consideration for the process.

Sincerely (on behalf of all authors),

Prof. Hyo Kang

Associate Professor
Department of Chemical Engineering (BK-21 Four Graduate Program)
Dong-A University
Busan 49315, Republic of Korea
Tel: +82 51 200 7720
Fax: +82 51 200 7728
E-mail [email protected]

Reviewer 2 Report

The water-drawing ability and LCST-type phase transition behavior of IL draw solutes composed of tetrabutylphosphonium cations and anions with different hydrophobicity  were described in this article, which is interesting and practically advantageous for application as FO draw solutes. I like this kind of study, yes, indeed, this paper deserves better journal and higher IF journal.

Author Response

Dear Editor at Membranes

We gratefully appreciate your kind reviewing and considering for publication in “Membranes”. We are submitting a revised manuscript (Membranes-2193805) entitled “Anion Effect on Forward Osmosis Performance of Tetrabutylphosphonium-based Draw Solute Having Lower Critical Solution Temperature”

We carefully read the reviewer’s comments and your e-mail. Reviewers gave us helpful comments on our manuscript. We think the reviewer’s opinion and suggestion is fairly reasonable. Therefore, we revised our manuscript taking the reviewer’s comments into consideration as follows. As you and the Reviewer suggested we modified some parts of the manuscript and the changes are shown in yellow texts. These changes are listed as follows:

Referee’s comments:

Referee: 2

Comments:

The water-drawing ability and LCST-type phase transition behavior of IL draw solutes composed of tetrabutylphosphonium cations and anions with different hydrophobicity  were described in this article, which is interesting and practically advantageous for application as FO draw solutes. I like this kind of study, yes, indeed, this paper deserves better journal and higher IF journal.

Answer:

We are much obliged for the comments suggested by Reviewer. Reviewer’s comments were very helpful to improve the quality of our manuscript.

We believe that now we answered all of the comments pointed out by the reviewers. I hope that now this paper is publishable in “Membranes”, one of the top journals in the field of membrane applications.

We also believe that this paper is also suitable for publication in “Membranes” from the following reasons.

  1. We prepared tetrabutylphosphonium-based ionic liquids (ILs), tetrabutylphosphonium benzenesulfonate ([P4444][BS]), tetrabutylphosphonium p-methylbenzenesulfonate ([P4444][MBS]), tetrabutylphosphonium p-ethylbenzenesulfonate ([P4444][EBS]), and tetrabutylphosphonium methanesulfonate ([P4444][MS]) to investigate their applicability as draw solute in forward osmosis (FO) process.

  1. In aqueous solutions, [P4444][MBS] and [P4444][EBS] show lower critical solution temperature (LCST) characteristics, which is essential for recovery of the draw solute after the FO permeation process. The LCSTs of 20 wt% aqueous solutions of [P4444][MBS] and [P4444][EBS] were approximately 36 and 25 ºC, respectively. These results signify that [P4444][MBS] and [P4444][EBS] aqueous solutions are efficient in terms of energy requirements for recovering the draw solute after FO process.

  1. The structure-property relationship is very important for designing the suitability of thermo-responsive ILs for draw solutions as well as the design of materials. Because LCSTs of thermo-responsive ILs affected by hydrophilicity and hydrophobicity of the cation or anion species are distinct and attractive characteristics in a narrow range between miscible and immiscible properties in solution. Therefore, we investigate the effect of change in the hydrophobic moiety of anions in ILs on the FO permeation and recovery behavior by varying anions with different hydrophobicity (benzenesulfonate anion ([BS]), para-position alkyl-substituted benzenesulfonate anions (p-methylbenzenesulfonate ([MBS]) and p-ethylbenzenesulfonate ([EBS])), and methanesulfonate anion ([MS])) in tetrabutylphosphonium ([P4444]+)-based ILs to design appropriate draw solutes. As a result, this study could contribute to the basic understanding of how the structure of the ions influences the drawing ability and recovery property of the ILs.

  1. We believe that our study makes a significant contribution to the literature because the thermo-responsive IL, which provides the thermo-responsive liquid-liquid phase separation property in the aqueous solutions was shown to be a promising draw solute that may breakthroughs, surpassing the limitations of technologies in existence such as high recovery temperature such as LCST, and trade-off relationship between osmotic pressure and the recovery temperature in the ionic liquid system.

In view of these achievements, we believe that our work represents a timely methodological advance and breakthrough in the field of membrane applications and thus is appropriate for a journal with the scope and wider readership of “Membranes”.

I hereby certify that this manuscript consists of original, unpublished work which is not under consideration for publication elsewhere.

We are excited to share our manuscript with you and look forward to hearing good news from you soon.

Thank you very much for your time and consideration for the process.

Sincerely (on behalf of all authors),

Prof. Hyo Kang

Associate Professor
Department of Chemical Engineering (BK-21 Four Graduate Program)
Dong-A University
Busan 49315, Republic of Korea
Tel: +82 51 200 7720
Fax: +82 51 200 7728
E-mail [email protected]

Reviewer 3 Report

Authors should discuss how the ILs can compete with typical draw solutes in terms of cost, availability, stability and safety.

Please discuss the safety level of using tet-rabutylphosphonium ([P4444]+)-based ILs as draw solute for FO process.

Line 115-116 - Hydration Technologies Inc. (HTI) already ceased business many years ago and how could the authors preserve the membranes for so many years and could still use it for experiments.

Besides, authors should demonstrate if there is any reverse draw solute from the draw solution containing ILs during FO process.

Figure 5 – I advise the authors to discuss the osmotic pressure in terms of bar or atm. This is the most commonly used unit for FO and PRO process. Besides, the discussion should be made by comparing the osmotic pressures of tet-rabutylphosphonium ([P4444]+)-based ILs with the widely used NaCl.

Section 3.4 – Extend the discussion by comparing the thermo-responsive property of tet-rabutylphosphonium ([P4444]+)-based ILs studied in this work with other relevant studies.

Section 3.5 – The performance evaluation of membrane is too simple. I strongly advise the authors to include additional experiments such as (a) stability of the membrane performance as a function of time and (b) the chemical/physical interaction of ILs with the membrane surface. The obtained results should be discussed fundamentally.

Author Response

Dear Editor at Membranes

We gratefully appreciate your kind reviewing and considering for publication in “Membranes”. We are submitting a revised manuscript (Membranes-2193805) entitled “Anion Effect on Forward Osmosis Performance of Tetrabutylphosphonium-based Draw Solute Having Lower Critical Solution Temperature”

We carefully read the reviewer’s comments and your e-mail. Reviewers gave us helpful comments on our manuscript. We think the reviewer’s opinion and suggestion is fairly reasonable. Therefore, we revised our manuscript taking the reviewer’s comments into consideration as follows. As you and the Reviewer suggested we modified some parts of the manuscript and the changes are shown in yellow texts. These changes are listed as follows:

Referee’s comments:

Referee: 3

Comments:

Authors should discuss how the ILs can compete with typical draw solutes in terms of cost, availability, stability and safety. Please discuss the safety level of using tet-rabutylphosphonium ([P4444]+)-based ILs as draw solute for FO process.

Answer:

We deeply thank you for the comment suggested by Reviewer and we revised and added the discussion regarding the cost, availability, stability and safety of the phosphonium based-ILs in the Introduction section, as “Among various ILs, phosphonium-based ILs have been reported to have more thermal and chemical stability [58–60]. The phosphonium-based ILs become attractive materials due to their competitive cost of synthesis and higher productivity at the industrial production level as well as their stability [61,62]. According to previous literature, the ILs based on phosphonium with shorter alkyl chain lengths have shown low toxicity toward several human microorganisms and they present the potential for bioprocessing applications [63]. Therefore, we have investigated the usability of tetrabutylphosphonium-based ILs in combination with benzenesulfonate which has a sulfonate group attached to the phenyl ring, a simple hydrophobic group. The hydrophobicity of benzenesulfonate anions can be tuned by attaching the extra methyl group to the phenyl ring. In addition, benzenesulfonate also has the advantage of low production cost and high thermal stability [64,65].” Added literatures are below. We strongly believe that Reviewer will satisfy our answer to this question.

[58] Maton, C.; De Vos, N.; Stevens, C.V. Ionic Liquid Thermal Stabilities: Decomposition Mechanisms and Analysis Tools. Chem. Soc. Rev. 2013, 42, 5963–5977.

[59] Bradaric, C.J.; Downard, A.; Kennedy, C.; Robertson, A.J.; Zhou, Y. Industrial Preparation of Phosphonium Ionic Liquids. Green Chem. 2003, 5, 143–152.

[60] Stalpaert, M.; Cirujano, F.G.; De Vos, D.E. Tetrabutylphosphonium Bromide Catalyzed Dehydration of Diols to Dienes and its Application in the Biobased Production of Butadiene. ACS Catalysis 2017, 7, 5802–5809.

[61] Khazalpour, S.; Yarie, M.; Kianpour, E.; Amani, A.; Asadabadi, S.; Seyf, J.Y.; Rezaeivala, M.; Azizian, S.; Zolfigol, M.A. Applications of Phosphonium-Based Ionic Liquids in Chemical Processes. J. Iran. Chem. Soc. 2020, 17, 1775–1917.

[62] Pena, C.A.; Soto, A.; Rodríguez, H. Tetrabutylphosphonium Acetate and its Eutectic Mixtures with Common-Cation Halides as Solvents for Carbon Dioxide Capture. Chem. Eng. J. 2021, 409, 128191.

[63] Mikkola, S.; Robciuc, A.; Lokajova, J.; Holding, A.J.; Lämmerhofer, M.; Kilpelainen, I.; Holopainen, J.M.; King, A.W.; Wiedmer, S.K. Impact of Amphiphilic Biomass-Dissolving Ionic Liquids on Biological Cells and Liposomes. Environ. Sci. Technol. 2015, 49, 1870–1878.

[64] Thomas, S.; Rayaroth, M.P.; Menacherry, S.P.M.; Aravind, U.K.; Aravindakumar, C.T. Sonochemical Degradation of Benzenesulfonic Acid in Aqueous Medium. Chemosphere 2020, 252, 126485.

[65] Masoudian, Z.; Salehi-Lisar, S.Y.; Norastehnia, A. Phytoremediation Potential of Azolla Filiculoides for Sodium Dodecyl Benzene Sulfonate (SDBS) Surfactant Considering some Physiological Responses, Effects of Operational Parameters and Biodegradation of Surfactant. Environ. Sci. Pollut. Res. 2020, 27, 20358–20369.

Line 115-116 - Hydration Technologies Inc. (HTI) already ceased business many years ago and how could the authors preserve the membranes for so many years and could still use it for experiments.

Answer:

The FO membrane purchased years ago has been stored in a sealed bag in a safer storage condition. We have re-probed the FO membrane with no ill effects. Until recently, the evaluation of the FO performance, such as water permeability of the various draw solution has been carried out using this FO membrane and the experimental data were reported in our previous articles which were published in several SCI journals. Likewise, this study includes the evaluation of FO performance and recovery process using this FO membrane and this result is reliable. We strongly believe that Reviewer will satisfy our answer to this question.

Besides, authors should demonstrate if there is any reverse draw solute from the draw solution containing ILs during FO process.

Answer:

We deeply thank you for the comment suggested by Reviewer and we measured the reverse solute flux of the [P4444][MBS] aqueous solutions at concentrations of 5, 10, 15, and 20 wt%. We revised and added this point in Materials and Methods section as “The reverse solute flux (Js, g m–2 h–1, gMH) represents the quantity of the permeated draw solute across the FO membrane to the feed solution. We calculated the reverse solute flux by comparing the conductivity difference of the feed solution before and after FO using the following equation (2):

                                                              (2)

Where C  is the concentration change,  is the variation in volume of the feed solution before and after FO, and  is the handing time of the experiment of the FO process.” and in Results and Discussion section as “The water flux and reverse solute flux of [P4444][MBS], which has a high osmolality potential among the thermo-responsive ILs, were measured to evaluate [P4444][MBS] as a draw solute in different orientations of the membrane placed between the two connected glass tubes; the glass tubes were filled with distilled water and [P4444][MBS] aqueous solution, respectively.” and “The reverse solute flux is the quantity of the permeated draw solute across the FO membrane to the feed water and was calculated by comparing the amount of total dissolved solids (TDS) of the distilled water (feed solution) before and after FO [94]. The reverse solute fluxes of the [P4444][MBS] aqueous solutions were 1.36, 1.47, 3.77, and 5.89 gMH in the AL-DS mode at 5, 11, 17, and 25 atm, respectively, and 1.14, 1.25, 2.45, and 2.67 gMH in the AL-FS mode, respectively, for the same condition.” We also revised Figure 8. The revised Figures and added literatures are below. We strongly believe that Reviewer will satisfy our answer about this question.

Figure 8. Water Fluxes and reverse solute fluxes of [P4444][MBS] aqueous solutions according to the osmotic pressure in AL-DS mode and AL-FS mode at a temperature of 25±1 oC during FO process.

[94] Ferby, M.; Zou, S.; He, Z. Reduction of reverse solute flux induced solute buildup in the feed solution of forward osmosis. Environ. Sci.: Water Res. Technol. 2020, 6, 423–435.

Figure 5 – I advise the authors to discuss the osmotic pressure in terms of bar or atm. This is the most commonly used unit for FO and PRO process. Besides, the discussion should be made by comparing the osmotic pressures of tet-rabutylphosphonium ([P4444]+)-based ILs with the widely used NaCl.

Answer:

We deeply thanks for comment suggested by Reviewer and we converted the osmolality values (mOsmol/kg) to the osmotic pressure (atm) using the van’t Hoff equation. We revised and added this point in Results and Discussion section as “To investigate the applicability as a draw solute, the osmolality values of the [P4444][MS], [P4444][BS], [P4444][MBS], and [P4444][EBS] aqueous solutions were measured for increasing IL concentrations from 5 to 20 wt% using the freezing point depression method. The osmolality values could be converted to the osmotic pressure via van’t Hoff equation using the temperature (T = 297 K) and density of the solution (density = 1g/ml), as depicted in Fig. 5. The osmotic pressures of the [P4444][MS], [P4444][BS], [P4444][MBS], and [P4444][EBS] aqueous solutions increase from 16.6, 13.7, 10.7, and 9.4 atm to 33.6, 33.3, 25.5, and 17.9 atm, respectively, when their concentration increases from 10 to 20 wt%.” We also revised Figure 5. The revised Figure is below. In addition, we added the discussion regarding the osmotic pressures of tetrabutylphosphonium ([P4444])-based ILs and the NaCl solutions in Results and Discussion section as “The [P4444][MS], [P4444][BS], and [P4444][MBS] aqueous solutions generate osmotic pressures of 33.6-25.5 atm at the concentration of 20 wt%, as compared to 25.5 atm generated by traditional inorganic salt especially NaCl with 3.5 wt%. Although osmotic pressures of the [P4444]-based ILs aqueous solutions are lower than NaCl, they exhibit potential as draw solution for brackish water treatment and food processing [85]. In addition, overall, the [P4444]-based ILs have not only good solubility and a high degree of dissociation at the concentration of 20 wt% but also thermal recovery properties.” We strongly believe that Reviewer will satisfy our answer about this question.

Figure 5. Osmotic pressures of [P4444][BS], [P4444][MBS], [P4444][EBS], and [P4444][MS] aqueous solutions according to the concentration of solution.

[85] Ou, R.; Wang, Y.; Wang, H.; Xu, T. Thermo-Sensitive Polyelectrolytes as Draw Solutions in Forward Osmosis Process. Desalination 2013, 318, 48–55.

Section 3.4 – Extend the discussion by comparing the thermo-responsive property of tet-rabutylphosphonium ([P4444]+)-based ILs studied in this work with other relevant studies.

Answer:

We deeply thank you for the comment suggested by Reviewer. We added the text and literatures about thermo-responsive property of the LCST-type ILs in Results and Discussion section as “The LCST behavior is attributed to the following equation, where ΔGmix is the mixing free energy, ΔHmix is the mixing enthalpy, and ΔSmix is the mixing entropy [87–89].

ΔGmix  ΔHmix TΔSmix                                                                       (3)

In the aqueous solution of IL system, the interaction between ions and water, such as hydrogen bonding play important role in the LCST-type phase behavior of the IL. The mixing enthalpy is negative at the low temperature because the hydrogen bonding makes a negative ΔSmix and a miscible phase between the component species, thus ΔGmix must be negative. Upon heating temperature above LCST, ΔGmix changes from negative to positive leading to phase separation due to the breaking of hydrogen bonding. In our system, the [P4444][MBS] and [P4444][EBS] aqueous solutions exhibit an LCST-type phase transition, as can be seen in Fig. 6. Below its LCST, the hydrogen interaction between water and the [P4444][MBS] and [P4444][EBS] is dominant. But above its LCST, the hydrogen interaction is weakened and hydrophobic interaction between [P4444]+ and benzenesulfonate derivatives anions ([MBS] and [EBS]) becomes dominant, thereby inducing the phase separation of [P4444][MBS] and [P4444][EBS]. In contrast, there were no noticeable changes in transmittance for the [P4444][MS] and [P4444][BS] aqueous solutions within the entire analyzed temperature range. Because the main factor affecting the LCST behavior of the ILs in water is the total hydrophobicity of the ILs [56]. When the IL composed of cation and anion are highly hydrophilic, IL is miscible in water and does not exhibit phase separation.” We strongly believe that Reviewer will satisfy our answer to this question.

[87] Zhao, C.; Ma, Z.; Zhu, X.X. Rational Design of Thermoresponsive Polymers in Aqueous Solutions: A Thermodynamics Map. Prog. Polym. Sci. 2019, 90, 269–291.

[88] Pasparakis, G.; Tsitsilianis, C. LCST Polymers: Thermoresponsive Nanostructured Assemblies Towards Bioapplications. Polymer 2020, 211, 123146.

[89] Kang, H.; Suich, D.E.; Davies, J.F.; Wilson, A.D.; Urban, J.J.; Kostecki, R. Molecular Insight into the Lower Critical Solution Temperature Transition of Aqueous Alkyl Phosphonium Benzene Sulfonates. Commun. Chem. 2019, 2, 51.

Section 3.5 – The performance evaluation of membrane is too simple. I strongly advise the authors to include additional experiments such as (a) stability of the membrane performance as a function of time and (b) the chemical/physical interaction of ILs with the membrane surface. The obtained results should be discussed fundamentally.

Answer:

We deeply thank you for the comment suggested by Reviewer. However, we could not obtain reliable results from the stability test of the membrane performance as a function of time due to the limitation of our custom-made FO system. We repeated the operation of the FO system four times using a 20 wt% aqueous solution of the [P4444][MBS] and added this point in Results and Discussion section, as “To explore the recyclable property of the [P4444][MBS], the FO performance of 20 wt% aqueous solutions of the [P4444][MBS] was repeated four times when the distilled water is used as the feed solution. The thermal treatment method used to obtain the recycled [P4444][MBS]. As shown in Fig. 9 (a) and (b), the osmotic pressure and LCST of the [P4444][MBS] were measured to confirm the recyclability of [P4444][MBS] at the fourth run. The osmotic pressure values at 4th run are almost the same as that of the pristine [P4444][MBS], while the LCST value slightly increases after the 4th run. These recycling results clearly show that [P4444][MBS] can be easily recycled with relatively low energy consumption without significant loss.” In addition, we measured the contact angles of the [P4444][MBS] aqueous solutions and distilled water on the surface of the active layer in the FO membrane. Because it is known that wettability is a macroscopic representation of the interfacial interaction between fluid and membrane surface and is usually characterized by the contact angle. This point was added in Results and Discussion section, as “The wettability of the membrane surface is a macroscopic representation of the interfacial interaction between fluid and membrane surface, and plays important role in the membrane performance [90–93]. The surface wetting of the membrane is usually characterized by the contact angle. The contact angles of the [P4444][MBS] aqueous solutions with various concentrations and distilled water on the surface of the active layer in the FO membrane were measured to understand the effect of the draw solution ([P4444][MBS] aqueous solution) on the wettability of the membrane. As shown in Fig. 7, the average contact angle of distilled water is lower than that of [P4444][MBS] aqueous solution at all concentration on the membrane surface and follow the trend: distilled water > 5 wt% [P4444][MBS] > 10 wt% [P4444][MBS] > 15 wt% [P4444][MBS] > 20 wt% [P4444][MBS]. The distilled water contact angle was 75°. The contact angles of the [P4444][MBS] aqueous solutions were 54, 49, 39, and 21°, respectively, at a concentration of 5, 10, 15, and 20 wt%, respectively. The contact angle of [P4444][MBS] aqueous solution on the membrane becomes smaller with increasing its concentration. This means that the wettability of the FO membrane was enhanced, with decreasing the contact angle of the [P4444][MBS] aqueous solution from 54° to 21°.” We revised and added the detailed information of contact angle measurements in Instrumentation section as “The measurement of the contact angle was performed using a Krüss DSA10 (KRÜSS Sci-entific Instruments Inc., Germany) contact angle analyzer equipped with drop shape analysis software after deposing the water and aqueous solution of ILs droplets on the FO membrane surface. The average volume of the droplets was 5 μL. The contact angles of each solution were determined four or more times.” We also added Figure 7 and Figure 9. The Figures and added literatures are below. We strongly believe that Reviewer will satisfy our answer about this question.

Figure 7. Contact angles of the [P4444][MBS] aqueous solution at the concentration of 0 (distilled water), 5, 10, 15, and 20 wt% on the FO membrane.

Figure 9. (a) Osmotic pressures and (b) LCST results using 20 wt% [P4444][MBS] aqueous solution as a draw solution and distilled water as a feed solution. From the 1st to the 4th run, the recovered [P4444][MBS] from the previous run was used.

[90] Wang, Z.; Elimelech, M.; Lin, S. Environmental Applications of Interfacial Materials with Special Wettability. Environ. Sci. Technol. 2016, 50, 2132–2150.

[91] Tiraferri, A.; Kang, Y.; Giannelis, E. P.; Elimelech, M. Highly Hydrophilic Thin-Film Composite Forward Osmosis Membranes Functionalized with Surface-Tailored Nanoparticles. ACS Appl. Mater. Interfaces 2012, 4, 5044–5053

[92] Gao, K.; Kearney, L.T.; Wang, R.; Howarter, J.A. Enhanced Wettability and Transport Control of Ultrafiltration and Reverse Osmosis Membranes with Grafted Polyelectrolytes. ACS Appl. Mater. Interfaces 2015, 7, 24839–24847.

[93] Ismail, M.F.; Islam, M.A.; Khorshidi, B.; Tehrani-Bagha, A.; Sadrzadeh, M. Surface Characterization of Thin-Film Composite Membranes using Contact Angle Technique: Review of Quantification Strategies and Applications. Adv. Colloid Interface Sci. 2022, 299, 102524.

We believe that now we answered all of the comments pointed out by the reviewers. I hope that now this paper is publishable in “Membranes”, one of the top journals in the field of membrane applications.

We also believe that this paper is also suitable for publication in “Membranes” from the following reasons.

  1. We prepared tetrabutylphosphonium-based ionic liquids (ILs), tetrabutylphosphonium benzenesulfonate ([P4444][BS]), tetrabutylphosphonium p-methylbenzenesulfonate ([P4444][MBS]), tetrabutylphosphonium p-ethylbenzenesulfonate ([P4444][EBS]), and tetrabutylphosphonium methanesulfonate ([P4444][MS]) to investigate their applicability as draw solute in forward osmosis (FO) process.

  1. In aqueous solutions, [P4444][MBS] and [P4444][EBS] show lower critical solution temperature (LCST) characteristics, which is essential for recovery of the draw solute after the FO permeation process. The LCSTs of 20 wt% aqueous solutions of [P4444][MBS] and [P4444][EBS] were approximately 36 and 25 ºC, respectively. These results signify that [P4444][MBS] and [P4444][EBS] aqueous solutions are efficient in terms of energy requirements for recovering the draw solute after FO process.

  1. The structure-property relationship is very important for designing the suitability of thermo-responsive ILs for draw solutions as well as the design of materials. Because LCSTs of thermo-responsive ILs affected by hydrophilicity and hydrophobicity of the cation or anion species are distinct and attractive characteristics in a narrow range between miscible and immiscible properties in solution. Therefore, we investigate the effect of change in the hydrophobic moiety of anions in ILs on the FO permeation and recovery behavior by varying anions with different hydrophobicity (benzenesulfonate anion ([BS]), para-position alkyl-substituted benzenesulfonate anions (p-methylbenzenesulfonate ([MBS]) and p-ethylbenzenesulfonate ([EBS])), and methanesulfonate anion ([MS])) in tetrabutylphosphonium ([P4444]+)-based ILs to design appropriate draw solutes. As a result, this study could contribute to the basic understanding of how the structure of the ions influences the drawing ability and recovery property of the ILs.

  1. We believe that our study makes a significant contribution to the literature because the thermo-responsive IL, which provides the thermo-responsive liquid-liquid phase separation property in the aqueous solutions was shown to be a promising draw solute that may breakthroughs, surpassing the limitations of technologies in existence such as high recovery temperature such as LCST, and trade-off relationship between osmotic pressure and the recovery temperature in the ionic liquid system.

In view of these achievements, we believe that our work represents a timely methodological advance and breakthrough in the field of membrane applications and thus is appropriate for a journal with the scope and wider readership of “Membranes”.

I hereby certify that this manuscript consists of original, unpublished work which is not under consideration for publication elsewhere.

We are excited to share our manuscript with you and look forward to hearing good news from you soon.

Thank you very much for your time and consideration for the process.

Sincerely (on behalf of all authors),

Prof. Hyo Kang

Associate Professor
Department of Chemical Engineering (BK-21 Four Graduate Program)
Dong-A University
Busan 49315, Republic of Korea
Tel: +82 51 200 7720
Fax: +82 51 200 7728
E-mail [email protected]

Round 2

Reviewer 1 Report

All of the reviewer comments have been addressed and all necessary revisions have been conducted.

Reviewer 3 Report

I'm now satisfied with the responses provided.